# Effect of Transition Metal Layer on Bending and Interfacial Properties of W/TiN/Ta-Laminated Composite

**DOI:** 10.3390/ma16062434

**Published:** 2023-03-18

**Authors:** Gaoyong Xu, Jili Cai, Ruoqi Wang, Ang Xu, Yifei Hu, Jilong Liu, Jinping Suo

**Affiliations:** State Key Laboratory of Material Processing and Die & Mould Technology, School of Materials Science and Engineering, Huazhong University of Science and Technology, Wuhan 430074, China

**Keywords:** W-based laminated composite, transition metal layer, bending property, interface

## Abstract

The widespread applications of W in the fusion reactor are limited by its low-temperature brittleness, recrystallization brittleness, and irradiation-induced brittleness. Many toughening methods were used to improve the brittleness of W, such as adding second-phase particles, adding W fibers, preparing laminated composite, and so on. Among these, preparing laminated W-based composites has been proven to effectively improve both the low-temperature and high-temperature toughness of W. In this study, W/M/TiN/Ta-laminated composites with transition metal layer (M) were synthesized through the spark plasma sintering (SPS) at three different temperatures. The effects of nano-scale (Ni, Ti, and Cr) and micron-scale (Ni, Ti, and V) transition layers on the bending and interfacial properties of the W/M/TiN/Ta composite were studied via an electron probe micro-analyzer (EPMA) and transmission electron microscope (TEM). Compared with W/TiN/Ta, the flexural strength and strain of W/Ni_nm_/TiN/Ta were increased by 25.6% and 17.6%, respectively. Ni, Ti, and V micron transition layers can improve the combination of the W–TiN interface and decrease the joining temperature. The micron V layer has the best strengthening effect. The flexural strength of W/V/TiN/Ta reached 1294 MPa, much higher than W/Ta’s 1041 MPa.

## 1. Introduction

Tungsten is viewed as one of the favorite plasma-facing materials (PFMs) in future fusion reactors [1,2,3]. However, tungsten’s widespread application is limited due to its high ductile-to-brittleness temperature, recrystallization brittleness, and irradiation-induced brittleness [4,5,6]. Preparing laminated composite has been proven an effective way to improve the low-temperature toughness of W [1,7,8,9,10,11]. Many transition metals have an excellent toughening effect on W. Ta is regarded as the most suitable metal due to its high melting point, plasticity, and infinite solid solution with W [6,11,12,13]. The efficiency toughening method to improve the high-temperature brittleness of W is adding second-phase particles, including many carbides, oxides, nitrides, etc., to nail grain boundaries and hinder dislocation movement [14,15,16,17]. Moreover, adding ceramic layers, such as TiN, into the W base composite can significantly improve the composite’s high-temperature strength [13]. However, present studies show that W/TiN/Ta-laminated composite plasticity will decrease, and interfacial cracking propensity will be increased due to the non-coherent interface between TiN and W [11]. Ti, V, Ni, and Ta were often used to toughen W in laminates or powder metallurgy materials [7,18,19].

This paper designed a new W/M/TiN/Ta-laminated structure, where M represents the transition metal layer. Thin transition metal layers were added between W and TiN to improve the interface bonding. The effects of nano-scale (Ni, Ti, and Cr) and micron-scale (Ni, Ti, and V) transition layers on the properties of composites and their mechanisms were studied. Firstly, the W/M/TiN/Ta-laminated composites, containing several transition layers of different thicknesses and types, were prepared via the spark plasma sintering (SPS) method. Secondly, three-point bending tests were carried out and analyzed. Then, electron probe micro-analysis (EPMA) and transmission electron microscope (TEM) was used to analyze the interface composition and structure of the laminated composites. Finally, the toughening mechanism of the transition layers on the composites was discussed.

## 2. Materials and Methods

The W and Ta disks used in this chapter are 100 µm thick, and the preparation of the TiN coating is the same as that of the literature [11], and the thickness is about 1 µm. The nano-scale transition Ni, Ti, and Cr layers were prepared through electron beam evaporation (EBE). The target materials’ purity is over 99.99 wt.%, and the substrates are W disks (Φ 30 mm × 0.1 mm). The W disks are soaked in acetone, isopropanol, and absolute ethanol for 5 min in order of ultrasound and then placed on a metal EBE station (Peva-600E, Taiwan Juchang Technology, Taiwan). The vacuum chamber is 4.0 × 10^−4^ Pa, and the thickness of the coating is ~65 nm.

Micron-sized Ni, Ti, and V are hot-rolled foils. The Ni and Ti foils, with a thickness of ~30 µm, were purchased from Qinghe County Runde Metal Material Co., Ltd. (Xingtai, Hebei Province). The V foil, with a thickness of ~60 µm, was purchased from Greetech Metal Materials Co., Ltd. (Xingtai, Hebei Province). The impurities content of different metal foils is listed in Table 1, Table 2 and Table 3, and the basic physical properties of the metal are shown in Table 4.

Figure 1 shows the SPS joining process. Three joining temperatures (1200 °C, 1400 °C, 1600 °C) were selected to study the effect of the transition metal layer and joining temperature on the properties of the laminated composites.

The morphology and microstructure of the composites were analyzed by EPMA-8050G (Shimadzu, Kyoto, Japan). The three-point bending tests were carried out at room temperature with specimen dimensions of 18 mm (l) × 2.5 mm (w) × 2.5 mm (h), a support span of 16 mm, and a loading speed of 0.3 mm∙min^−1^ by a universal material testing machine (Zwick Z020, Zwick/Roell, Ulm, Germany). TEM (FEI Tecnai G2 F30, Hillsboro, OR, USA) analysis was carried out to observe the microstructure of the composites at high resolution. The TEM specimen was cut from the composites by a focused ion beam device (FEI Helios NanoLab 600i, Hillsboro, OR, USA). Then the TEM specimen was treated by ion milling into 10–50 nm in thickness. The thin area was observed and analyzed by TEM.

## 3. Results

### 3.1. Bending Properties of Nano-Scale Transition Layers Enhanced W/M_nm_/TiN/Ta-Laminated Composites

Two joining temperatures of 1200 °C and 1400 °C were carried out to prepare W/M_nm_/TiN/Ta-laminated composites, but the W and Ta disks failed to sinter as a bulk at 1200 °C.

Figure 2 shows the three-point bending stress-displacement curve of the composite material containing Ni, Ti, and Cr nano-scale transition layer joined at 1400 °C. Table 5 shows the strength, plasticity, and integration of W/M_nm_/TiN/Ta. The integration of the stress-deformation curve represents the relative value of the energy absorbed by the material during the fracture process. Only the Ni_nm_ layer significantly enhanced the flexural properties of the composite. Compared with W/TiN/Ta, the flexural strength and strain of W/Ni_nm_/TiN/Ta were increased by 25.6% and 17.6%, respectively. The Ti_nm_ layer does not affect the flexural properties, and the Cr_nm_ layer even reduces the plasticity of the composite.

Figure 3 shows the element distribution near the interface of W/Ta and W/TiN/Ta-laminated composites joined at 1400 °C. The width of the W/Ta interface is about 1.5 µm, which conforms to the composition distribution characteristics of a continuous solid solution. The W–TiN interface is about 0.7 µm, and there is almost no Ta trans–TiN diffusion due to the lower joining temperature.

Figure 4 shows the element line scanning results near the interface of W/M_nm_/TiN/Ta-laminated composites joined at 1400 °C, where M_nm_ represents nano-scale Ni, Ti, and Cr layers. Compared with W/TiN/Ta in Figure 3d, more Ta diffused across the TiN to W–TiN interface of W/M_nm_/TiN/Ta composites, especially for W/Ni_nm_/TiN/Ta. The diffusion distance of Ta in W/Ni_nm_/TiN/Ta reaches ~5 µm. The distance is ~1 µm for W/Ti_nm_/TiN/Ta and W/Cr_nm_/TiN/Ta. The difference in Ta diffusion among the three composites relates to Ni, Ti, and Cr melting points. The transition metal with a lower melting point had a more substantial promotion effect on the diffusion of Ta.

Figure 5 shows the three-point bending crack distribution of W/M_nm_/TiN/Ta-laminated composites joined at 1400 °C. W/Ni_nm_/TiN/Ta interfacial cracking decreased compared with W/TiN/Ta (Figure 5a,b). The cracking conditions of W/Ti_nm_/TiN/Ta and W/Cr_nm_/TiN/Ta are the same as those of W/TiN/Ta, and the interface cracking of the latter is slightly more severe (Figure 5c–f). The interface strength contributes to the bending strength and is also related to the energy consumed to form a new interface during the fracture process. The higher interface strength usually results in more fracture energy consumption.

### 3.2. Bending Properties of Micro-Scale Transition Layers Enhanced W/M/TiN/Ta-Laminated Composites

Figure 6 and Table 6 present the three-point bending properties of W/M/TiN/Ta composites with different transition metal layers joined at 1200 °C. Since W/TiN/Ta failed to join as a bulk at 1200 °C, only W/Ta is listed for comparison. All Ni, Ti, and V micron transition layers can improve the combination of the W–TiN interface and reduce the sintering temperature. Compared with the W–Ta’s bending properties, the W/Ni/TiN/Ta’s performance decreased significantly. Furthermore, W/Ti/TiN/Ta only had an apparent loss in strength. Though the plasticity of W/V/TiN/Ta almost did not change, the bending strength and integration are greatly improved.

Figure 7 shows the element line scanning results of W/M/TiN/Ta near the interface joined at 1200 °C. A large amount of molten Ni was extruded from the mold after joining. Most of the Ni diffused across TiN into Ta in W/Ni/TiN/Ta composites, and the diffusion distance reaches ~2 μm (Figure 7a). The diffusion of Ti led to the width of the TiN layer broadening to ~5 µm. Moreover, the diffusion distance of Ti across TiN reached ~19 μm (Figure 7b). The diffusion of V was hindered by TiN layers (Figure 7c).

According to the phase diagram, Ni and W have a variety of intermetallic compounds such as Ni_4_W, NiW, NiW_2_, etc. [20]. V and W can form a continuous solid solution [21]. Ni and Ta have various intermetallic compounds, such as Ni_x_Ta (x = 8, 3, 2, 1, 0.5) [22]. From the relative content distribution of Ni and Ta in Figure 7a, the intermetallic compound should mainly consist of NiTa and NiTa_2_.

Figure 8 shows the three-point bending crack distribution of W/Ni/TiN/Ta, W/Ti/TiN/Ta, and W/V/TiN/Ta-laminated composites joined at 1200 °C. No evident interfacial debonding was found, indicating that the three transition metal layers reduced the cracking propensity of the W–TiN interface (Figure 8a,c,e). Many microcracks appeared near the interface of the W/Ni/TiN/Ta-laminated composite after three-point bending (Figure 8b). The TiW, TiTa solid solutions, and the TiN layer formed a sandwich structure, which shows a brittle multi-crack fracture mode after the three-point bending test (Figure 8c). The TiTa/TiN/TiW sandwich structure presented a good transition with Ta layers (Figure 8d). However, the hindering effect of the TiN interface on the movement dislocations will be weakened. Bending crack deflection was more prominent for W/V/TiN/Ta. Few transverse cracks appeared in the W matrix, which indicated the V–W interface was bonded well.

Figure 9 shows the element profile of the W/Ni/TiN/Ta-laminated composite joined at 1200 °C. Ni and Ta formed needle-like structures. The structure should be a variety of Ni and Ta intermetallic compounds according to their phase diagram. The joining temperature closer to the melting point of Ni makes Ni highly active and easy to react with Ta, but there is no such obvious reaction between Ni and W [23,24]. The melting points of the intermetallic compounds between Ni and W are higher than that of Ni_x_Ta. Furthermore, Ni_2_Ta has the lowest melting point, 1330 °C.

When the temperature is above 1250 °C, W and Ti can form a continuous solid solution [25]. The actual internal temperature will be slightly higher than the set temperature due to the interior heating characteristics of SPS. Therefore, Ti will precipitate from the W–Ti solid solution when the temperature decreases [26].

Figure 10 and Table 7 present three-point bending properties of W/Ti/TiN/Ta and W/V/TiN/Ta-laminated composites as SPSed at 1400 °C and 1600 °C, respectively. W/Ta has an enormous strain and integration (689.4 kJ/m^2^) after joining at 1400 °C. W/V/TiN/Ta has the best bending strength. Compared with the W/Ta (733 MPa), the bending strength of W/TiN/Ta was increased by 21%. The bending strength and strain of W/V/TiN/Ta joined at 1600 °C further decreased but were still higher than that of W/TiN/Ta.

Figure 11 shows the size distribution of W grains in W/Ta joined at 1200 °C and 1400 °C. With the joining temperature increasing, the average size of W grains grew from 5.8 µm to 15.8 µm. Complete recrystallization of Ta occurs when the annealing temperature is over 1200 °C [27]. According to the Hall–Petch relation [28,29], grain growth should be the main reason for the decrease in strength of W/Ta-laminated composites. The plasticity of Ta was improved a lot due to the recovery of texture and dislocation.

Figure 12 shows the element profiles near the interface of W/Ti/TiN/Ta and W/V/TiN/Ta. In W/Ti/TiN/Ta, the diffusion distance of Ti elements into Ta is ~25 µm, and the TiN width is about 5 µm, forming a TiTa–TiN–TiW three-layer structure similar to that sintered at 1200 °C, but the width increases to about 45 µm. At the same time, the contents of Ta and W in TiTa and TiW solid solutions also increased significantly, indicating that the sintering temperature accelerated the diffusion **(**Figure 12b). In W/V/TiN/Ta (1400 °C), the diffusion distance of the V element increased by 5~10 µm compared with 1200 °C. At the same time, along with the dissolution of TiN, the Ti element mainly diffused into V, while the N element diffused more evenly to both sides (Figure 12d). In W/V/TiN/Ta (1600 °C), the diffusion distance of V reached ~20 µm, and the TiN layer wholly dissolved into the matrix. The mutual diffusion between Ta and V became more severe due to the dissolution of the TiN layer (Figure 12f).

Figure 13 shows the three-point bending cracks of W/Ti/TiN/Ta and W/V/TiN/Ta-laminated composites as SPSed at different joining temperatures. W/Ti/TiN/Ta (joined at 1400 °C) presented a multi-crack propagation pattern. The TiTa–TiN–TiW structure showed no signs of plastic deformation (Figure 13a,b). The W/V/TiN/Ta (1400 °C) cracks mainly propagate longitudinally, and the deflection of the cracks was tiny. The transverse cracks are primarily near the W–V interface (Figure 13c,d). With the increase of sintering temperature, the width of the VTa solid solution zone in W/V/TiN/Ta (1600 °C) increases. According to the crack distribution, crack deflection mainly occurs in the middle of the W layer, and all surfaces have no cracking propensity. However, the number of interfaces decreased due to the complete dissolution of the TiN layer. Under these factors, the strength and plasticity of the material decreased significantly (Figure 13e,f).

## 4. Discussion

Cracks did not extend from the Ta–Ti interface (Figure 8c,d and Figure 13a,b), indicating that the Ta–Ti interface was well bonded at 1200 °C and 1400 °C. Based on the Ta–V binary phase diagram [30,31], TaV will become a continuous solid solution over 1310 °C. The laves phase V_2_Ta with an FCC structure can be formed between 800 °C and 1310 °C, and laves phase is brittle at room temperature. There is a big chance for the formation of V_2_Ta under the three joining conditions. Due to the presence of the TiN layer, no evident diffusion of V and Ta was observed at 1200 °C (Figure 7c). So, the content of V_2_Ta should not be enough to affect the mechanical properties of the composite. The TiN layer started to dissolve into the matrix when the joining temperature reached 1400 °C. Then V_2_Ta was formed during the cooling stage (Figure 12d,f). As the joining temperature reached 1600 °C, the width of V and Ta solid solution increased to ~80 µm, and the V_2_Ta content should increase significantly. The VTa solid solution layer presented prominent brittle fracture characteristics at room temperature (Figure 13e,f). Therefore, the bending properties were affected by the solid solution strengthening effect of N elements and influenced by the brittle laves phase V_2_Ta.

Figure 14 shows the high-angle annular dark-field (HAADF) image and energy dispersive spectroscopy (EDS) results of W/Ti/TiN/Ta-laminated composites. Two interfaces were observed on both sides of the TiN_x_ phase (Figure 14a). The left phase was denoted as TiTa, mainly consisting of Ti and Ta. Similarly, the right one is marked as TiW (Figure 14c–f).

Figure 15 shows the high-resolution (HR)-TEM image of W/Ti/TiN/Ta composites and the orientation of the selected electron diffraction interface. Both TiTa and TiN_x_ were hexagonal close-packing (HCP) structures with incident crystal axes [0001] and [01$\overset{-}{1}$1], respectively (Figure 15a,b). The content of Ti in TiTa is nearly eight times as much as Ta’s in the atomic ratio (Figure 7b). Although the TiTa continuous solid solution formed during the joining process, β→α transformation of Ti will occur during the cooling process. As the width of TiN increases to ~3.5 µm (Figure 14a), TiN_x_ cannot maintain FCC structure and transforms into an HCP structure. Both TiTa and TiN_x_ have an HCP structure mainly consisting of Ti. Theoretically, they can form a nearly coherent interface. According to selected electron diffraction results, TiW and TiN_x_ are HCP structures. The electron beam incident along the crystal band axis [1$\overset{-}{1}$1] presents diffraction spots similar to precipitated phases, with one–two spots around the main diffraction spots. These precipitated phases may be W precipitated by Ti during the phase transition process. The TiW–TiN_x_ interface presents a nearly complete coherent relationship through high-resolution images (Figure 15d).

Figure 16 shows the HAADF image and EDS results of the W/V/TiN/Ta-laminated composite as SPSed at 1200 °C. The HAADF image shows a three-layer structure, the Ta–TiN–V (Figure 16a,b). The V element diffused to the Ta layer, and the N element diffused to both sides but more to the V layer (Figure 16c–f).

Figure 17 shows the HR-TEM image and the orientation relationship of the interface of W/V/TiN/Ta as SPSed at 1200 °C. Table 8 lists mismatch values (δ) and interplanar crystal spacing (D) values parallel crystal planes of W/V/TiN/Ta. According to the selected electron diffraction results, Ta is the BCC structure, and TiN is the FCC structure. At the Ta–TiN interface, crystal planes, (01$\overset{-}{1}$) Ta and (200) TiN, are approximately parallel. The crystal plane spacing is measured as 0.2338 nm and 0.2119 nm, respectively. The mismatch degree is ~9.8%, so this interface belongs to the semi-coherent interface (Figure 17a–c). There is also a group of parallel crystal planes ($\overset{-}{2}$20) V or ($\overset{-}{1}$10) V and (220) TiN, whose crystal plane spacing can be measured as 0.2143 nm, 0.1071 nm, and 0.1498 nm, respectively. The mismatch between ($\overset{-}{2}$20) V and (220) TiN is about 35.4%, and the mismatch between ($\overset{-}{1}$10) V and (220) TiN is about 33.2%. Therefore, the V–TiN interface can be considered a non-coherent interface. The deflection of cracks is mainly along the V–TiN interface and can be due to the non-coherent V–TiN interface being more likely to produce stress concentration. However, the alternately soft and hard phase structure alleviated the stress concentration between W and TiN.

Figure 18 shows the HAADF image and EDS scanning results of W/V/TiN/Ta-laminated composites as SPSed at 1400 °C. The TiN layer is no longer visible in the HAADF image. Many Ti and N elements diffused to Ta and V layers.

Figure 19 shows the HR-TEM image and the interfacial orientation relationship of W/V/TiN/Ta composites as SPSed at 1400 °C. Ta and VTa solid solutions are still the BCC structure despite the solid solution of a small amount of Ti and N elements (Figure 19a,b). The Ta–V interface is nearly entirely coherent. V and Ta can form continuous solid solutions. Although TiN has been dissolved, most Ti and N elements, especially Ti, are enriched near the interface. The diffraction spots at the Ta–V interface mostly coincided, indicating that the interface’s lattice mismatch value is small and the Ta–V interface is nearly coherent. At the same time, the TiN layer began to dissolve. The Ti and N concentrated around the V–Ta interface hindered dislocation movement by solution strengthening effect. Although the hindering impact was weaker than the TiN layer, it maintained high strength and plasticity. Compared with W/TiN/Ta, the integration decreased significantly due to the dissolution of TiN. At the same time, dissolved Ti and N elements reduced the plasticity of the V and Ta matrix.

## 5. Conclusions

In this paper, transition metal layers were used to construct a coherent or semi-coherent M-TiN interface to improve the toughness of the W/TiN/Ta composite. The effects of transition layers with nanometers (Ni, Ti, and Cr)/microns (Ni, Ti, and V) thickness and joining temperature on the bending and interfacial properties. The main conclusions are summarized as follows:The nano-transition layers (Ni, Ti, V) have a pronounced promoting effect on the diffusion of the Ta element across the TiN layer. Moreover, more diffusion of Ta was observed for transition metal layers with a lower melting point;The W/Ni_nm_/TiN/Ta’s bending strength and plasticity were increased to 1150 MPa and 1.1%, respectively. Nano Cr and Ti layers have nearly no impact on the properties of composites because they have little effect on Ta’s diffusion;The micron Ni, Ti, and V transition layers can reduce the composite’s preparation temperature and inhibit the interface’s cracking propensity. Adding V layers improved the interface bonding of the W/V/TiN/Ta composites. Although the interfacial debonding was reduced, the performance deterioration of the Ta layers decreased the mechanical properties of W/Ni/TiN/Ta and W/Ti/TiN/Ta;W/V/TiN/Ta has the best flexural strength (1294 MPa) and integration (419.3 kJ∙m^−2^) joined at 1200 °C. With the joining temperature increasing, the bending property of W/V/TiN/Ta decreased because of TiN’s dissolution in the V layers, leading to easier interface failure. At the same time, V_2_Ta decreased the toughening effect of V layers on the composite;The W/M/TiN/Ta-laminated composites could be another choice for PFMs in the fusion reactor, liner materials, and ultra-high temperature structural materials in aerospace fields.

## Figures and Tables

**Figure 1 materials-16-02434-f001:**
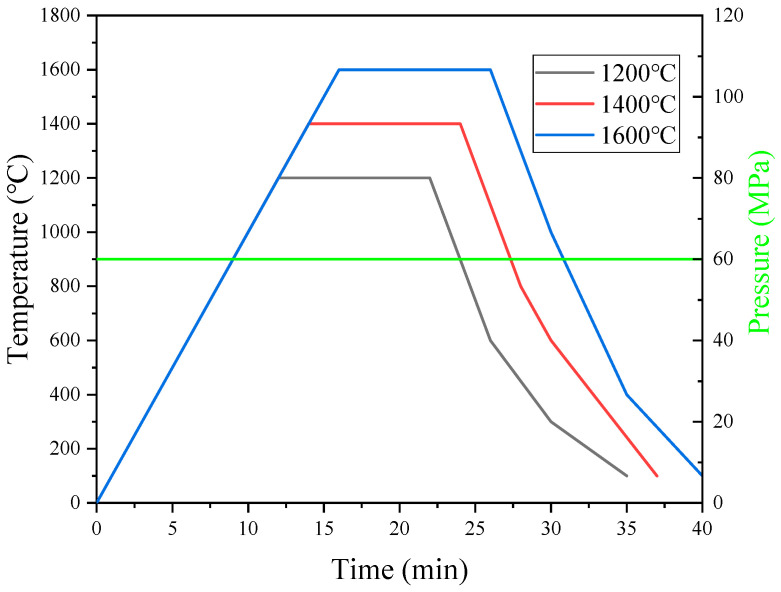
Curves of SPS sintering process.

**Figure 2 materials-16-02434-f002:**
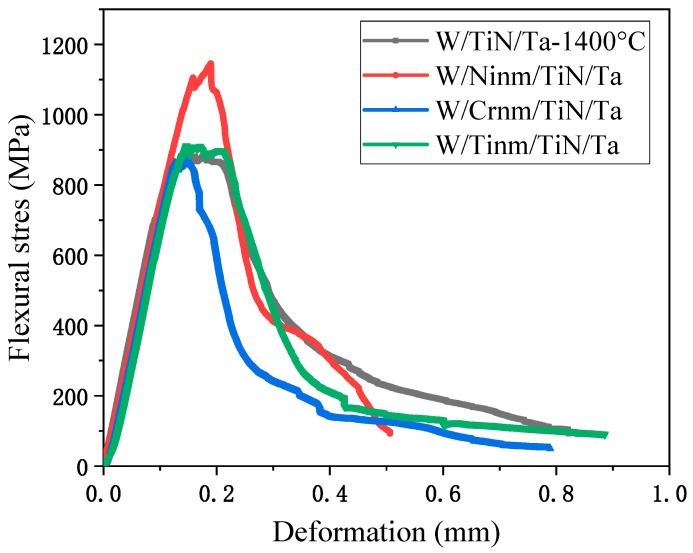
Three-point bending stress-displacement curves of different transition layer composites joined at 1400 °C.

**Figure 3 materials-16-02434-f003:**
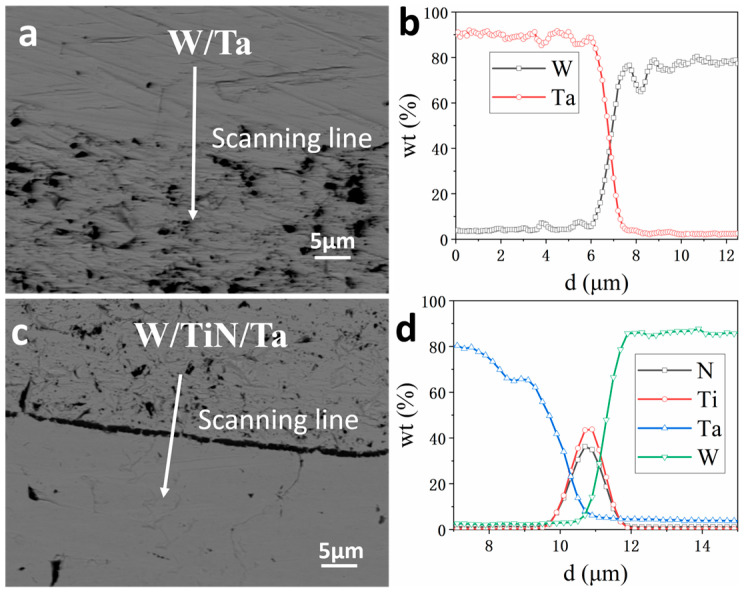
Element line-scan results of W/Ta and W/TiN/Ta-laminated composites joined at 1400 °C: (**a**,**b**) W/Ta, (**c**,**d**) W/TiN/Ta.

**Figure 4 materials-16-02434-f004:**
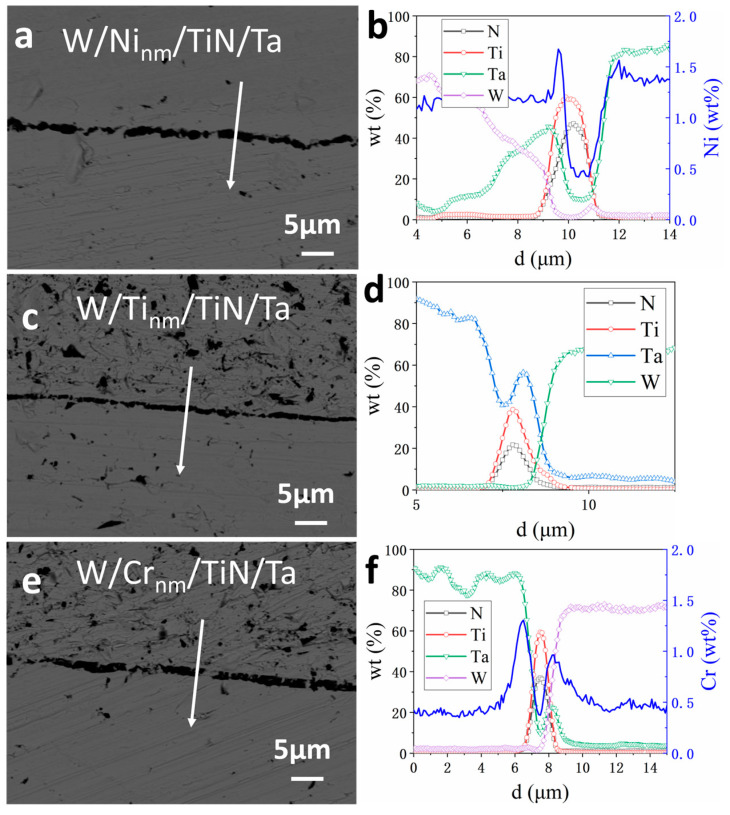
Element distribution near the interface of W/M_nm_/TiN/Ta-laminated composites joined at 1400 °C: (**a**,**b**) W/Ni_nm_/TiN/Ta, (**c**,**d**) W/Ti_nm_/TiN/Ta, (**e**,**f**) W/Cr_nm_/TiN/Ta.

**Figure 5 materials-16-02434-f005:**
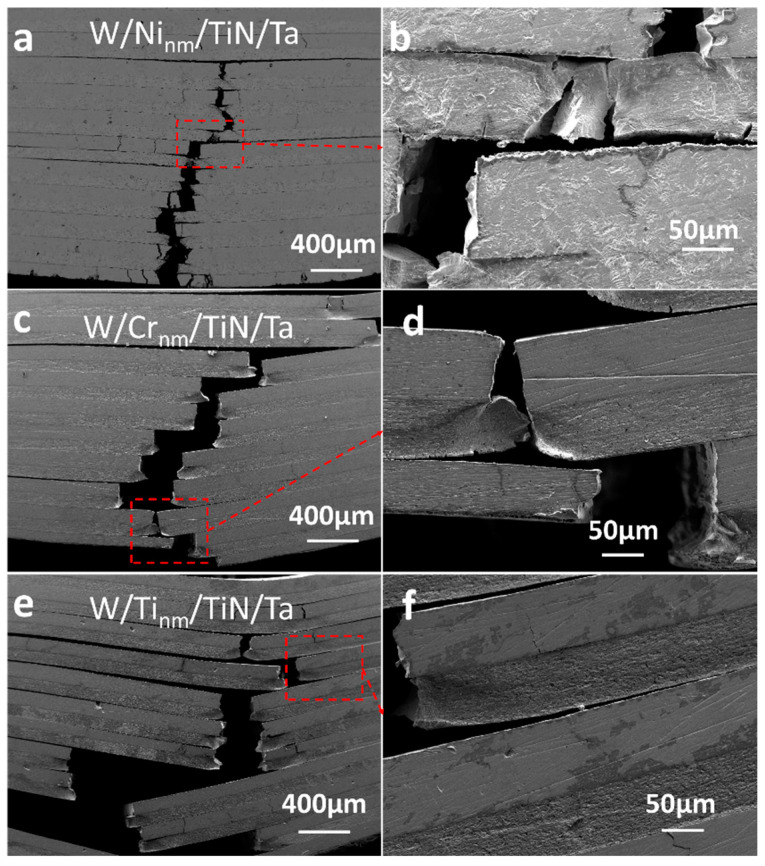
Three-point bending crack distributions of W/M_nm_/TiN/Ta-laminated composites with different transition metal layers joined at 1400 °C: (**a**,**b**) W/Ni_nm_/TiN/Ta, (**c**,**d**) W/Ti_nm_/TiN/Ta, and (**e**,**f**) W/Cr_nm_/TiN/Ta.

**Figure 6 materials-16-02434-f006:**
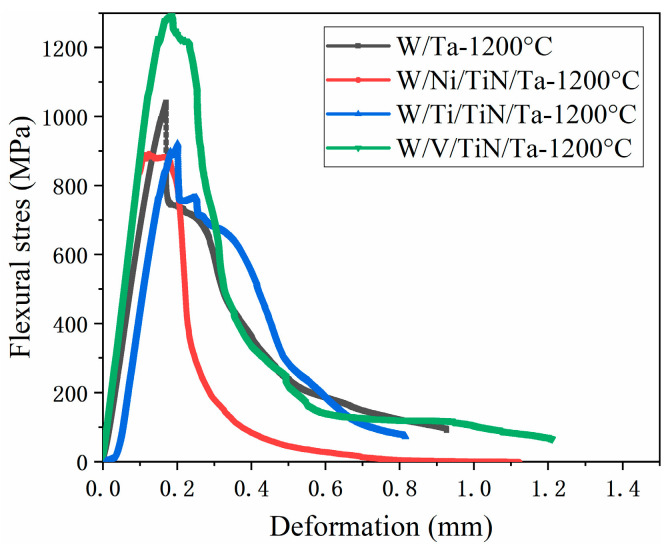
Three-point bending stress-deformation curves of different micron transition layer toughened composites joined at 1200 °C.

**Figure 7 materials-16-02434-f007:**
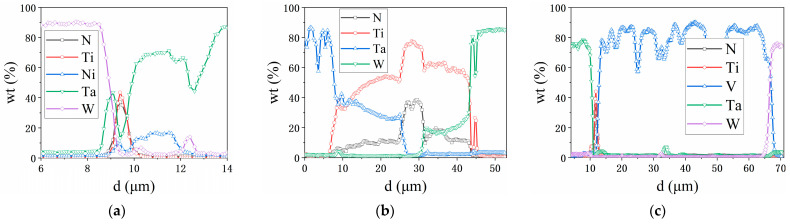
Element distribution near the interface of W/M/TiN/Ta-laminated composites with different transition metal layers joined at 1200 °C: (**a**) W/Ni/TiN/Ta; (**b**) W/Ti/TiN/Ta; and (**c**) W/V/TiN/Ta.

**Figure 8 materials-16-02434-f008:**
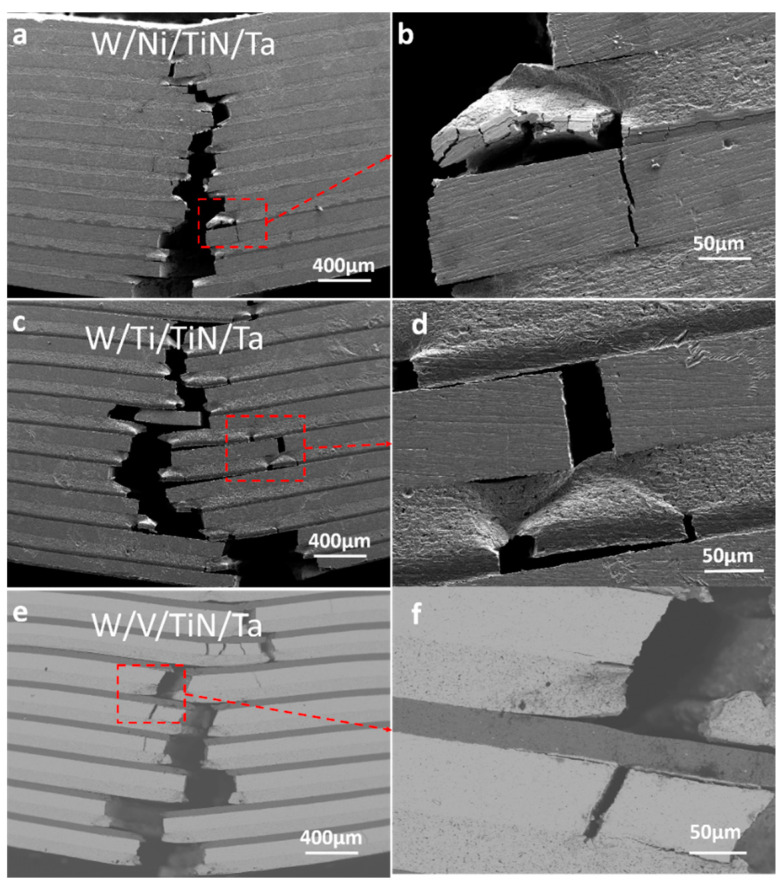
Three-point bending crack distribution of W/M/TiN/Ta-laminated composites with different transition metal layers joined at 1200 °C: (**a**,**b**) W/Ni/TiN/Ta; (**c**,**d**) W/Ti/TiN/Ta; and (**e**,**f**) W/V/TiN/Ta.

**Figure 9 materials-16-02434-f009:**
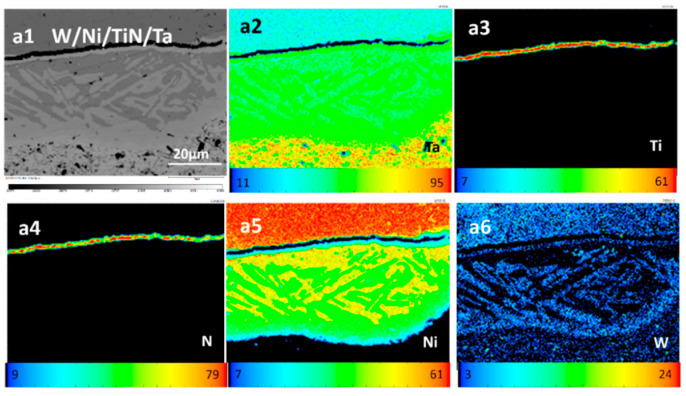
Interface composition distribution of W/Ni/TiN/Ta-laminated composite joined at 1200 °C: (**a1**) morphology; (**a2**) Ta; (**a3**) Ti; (**a4**) N; (**a5**) Ni; and (**a6**) W.

**Figure 10 materials-16-02434-f010:**
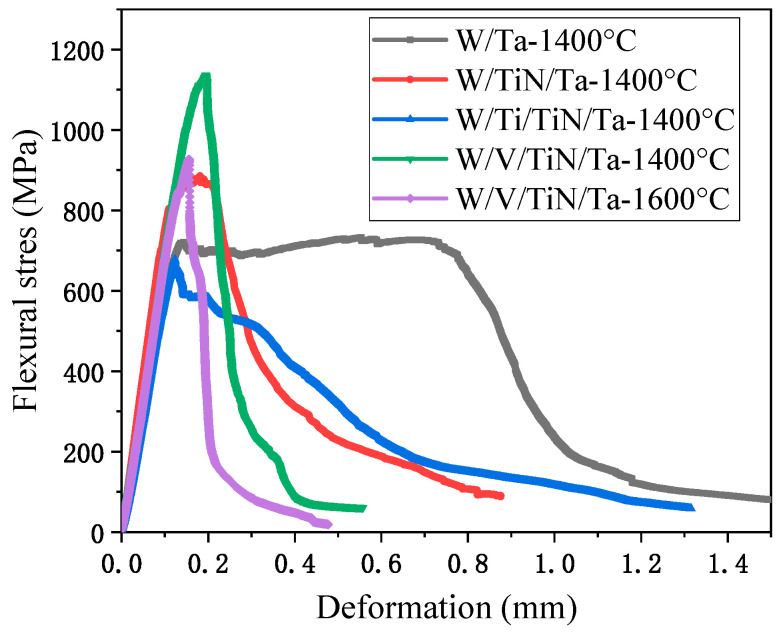
Three-point bending strength-displacement curves of W/M/TiN/Ta-laminated composites joined at 1400 °C and 1600 °C.

**Figure 11 materials-16-02434-f011:**
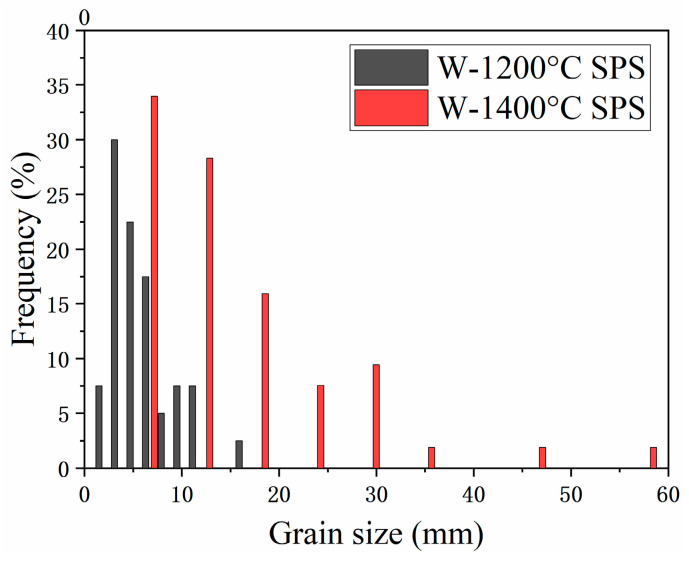
The size distribution of W grains in the W/Ta-laminated composite joined at 1200 °C.

**Figure 12 materials-16-02434-f012:**
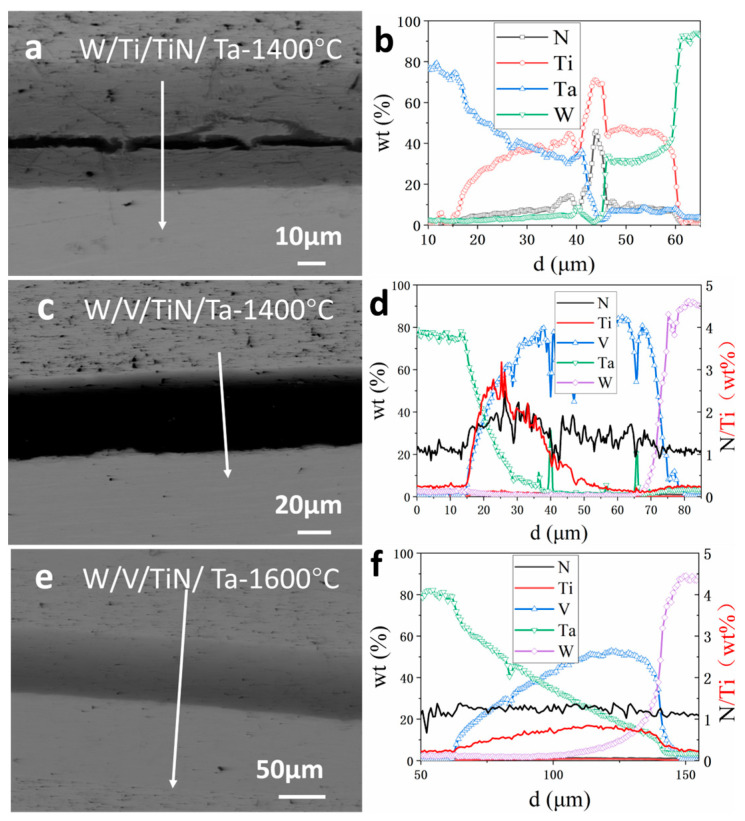
Element distribution near the interface of W/M/TiN/Ta-laminated composites: (**a**,**b**) W/Ti/TiN/Ta (joined at 1400 °C); (**c**,**d**) W/V/TiN/Ta (joined at 1400 °C); and (**e**,**f**) W/V/TiN/Ta (joined at 1600 °C).

**Figure 13 materials-16-02434-f013:**
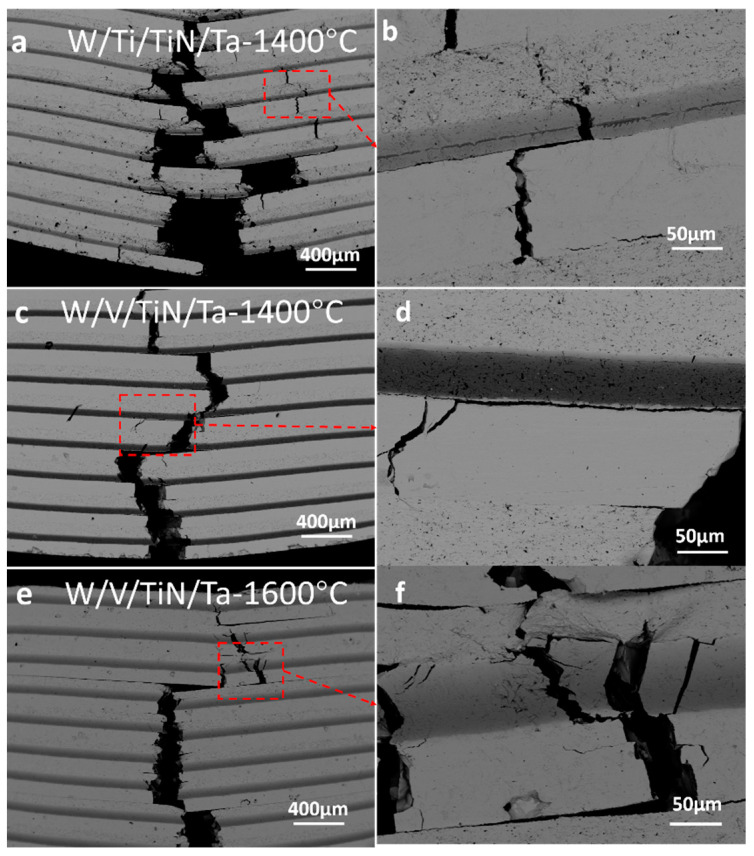
Three-point bending crack distribution of W/M/TiN/Ta-laminated composites with different transition metal layers after SPS: (**a**,**b**) W/Ti/TiN/Ta (1400 °C); (**c**,**d**) W/V/TiN/Ta (1400 °C); and (**e**,**f**) W/V/TiN/Ta (1600 °C).

**Figure 14 materials-16-02434-f014:**
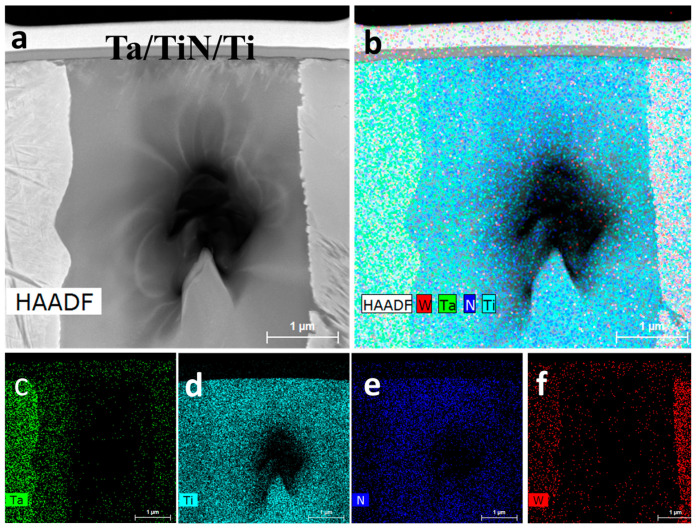
W/Ti/TiN/Ta-laminated composite joined at 1200 °C: (**a**) HAADF image; EDS mapping profile of (**b**) W, Ta, Ti, N, (**c**) Ta, (**d**) Ti, (**e**) N, and (**f**) W.

**Figure 15 materials-16-02434-f015:**
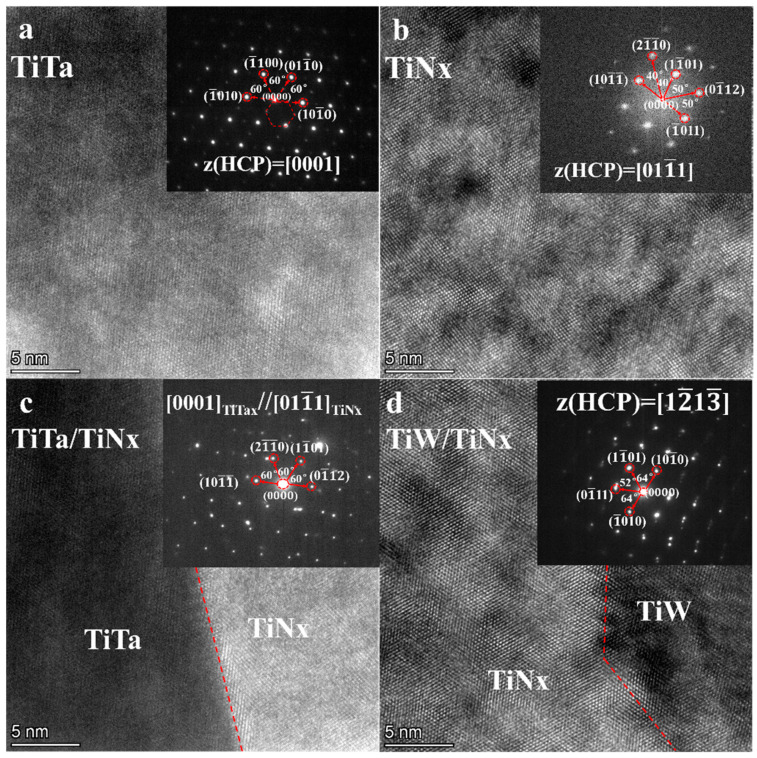
HR-TEM images of W/Ti/TiN/Ta composites after 1200 °C joining and the orientation relationships of the interface: (**a**) TiTa; (**b**) TiNx; (**c**) TiTa–TiNx interface; and (**d**) TiW–TiNx interface.

**Figure 16 materials-16-02434-f016:**
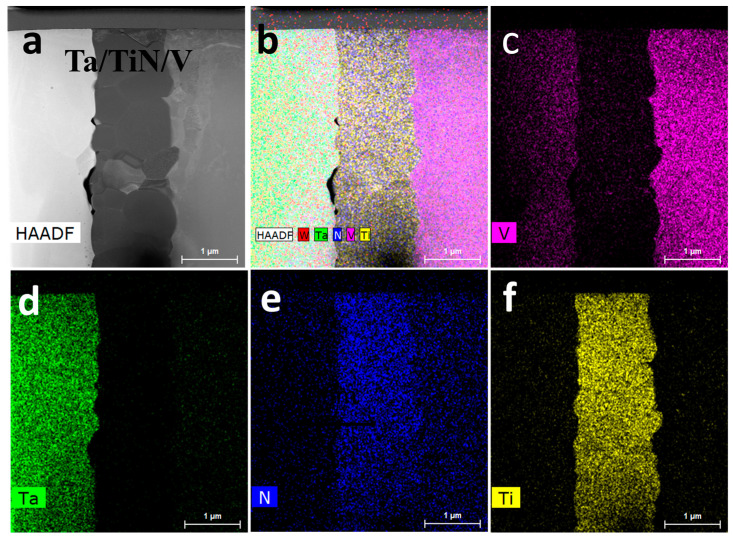
W/V/TiN/Ta-laminated composite joined at 1200 °C: (**a**) HAADF image, EDS mapping profile of (**b**) W, Ta, Ti, N, V, (**c**) V, (**d**) Ta, (**e**) N, and (**f**) Ti.

**Figure 17 materials-16-02434-f017:**
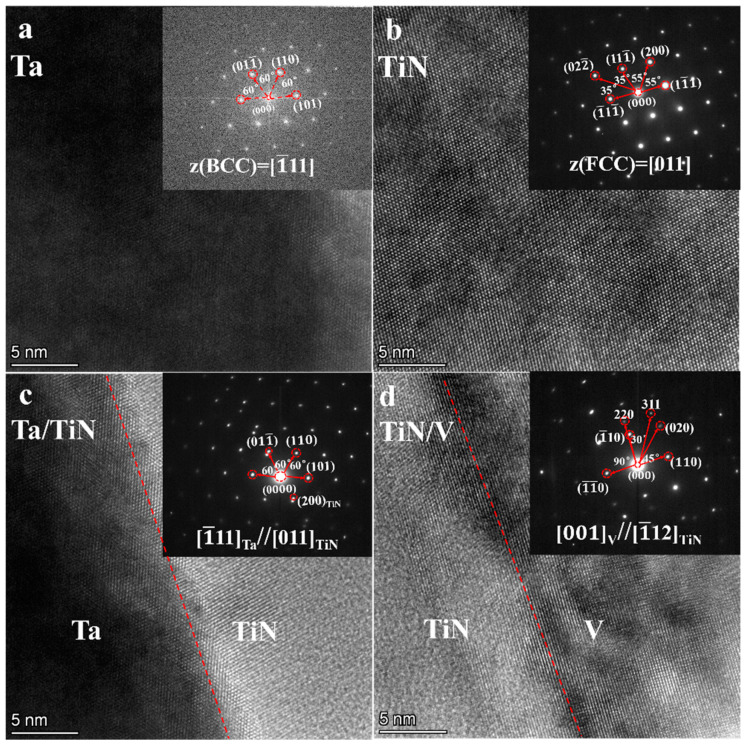
HR-TEM images and the orientation relationships of the interface of W/V/TiN/Ta composites as SPSed at 1200 °C: (**a**) Ta; (**b**,**d**) TiN; (**c**) Ta–TiN interface; and (**d**) V–TiN interface.

**Figure 18 materials-16-02434-f018:**
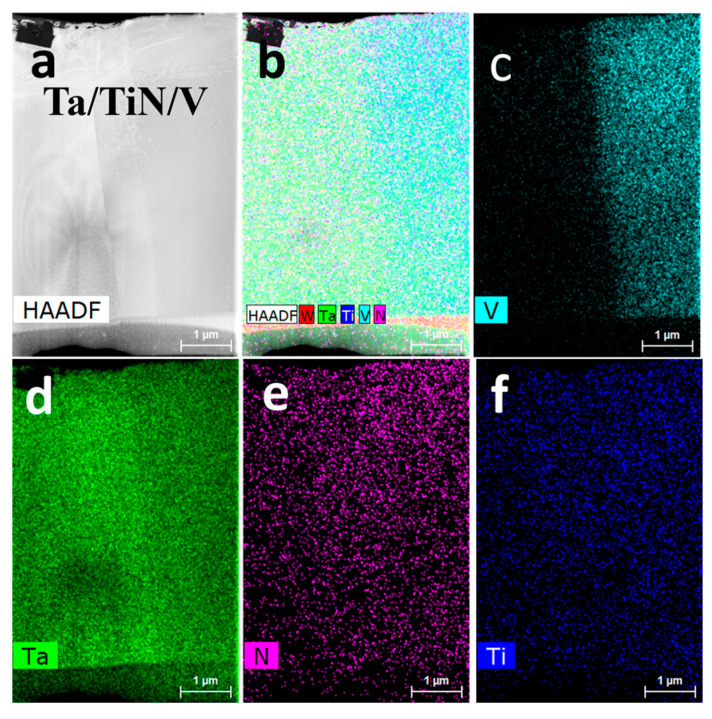
W/V/TiN/Ta-laminated composite joined at 1400 °C: (**a**) HAADF image; EDS mapping profile of (**b**) W, Ta, Ti, N, V, (**c**) V, (**d**) Ta, (**e**) N, (**f**) Ti.

**Figure 19 materials-16-02434-f019:**
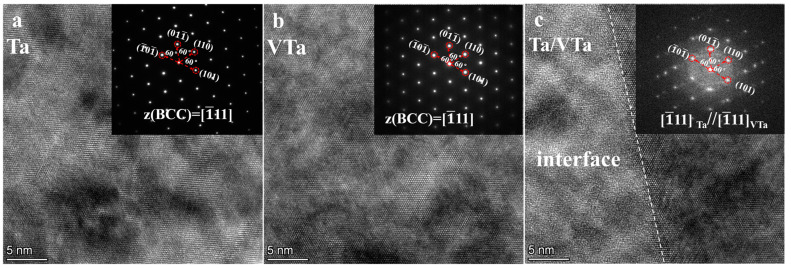
Orientation relationships between HR-TEM image and interface in W/V/TiN/Ta composite joined at 1400 °C: (**a**) Ta; (**b**) VTa; (**c**) Ta–VTa interface.

**Table 1 materials-16-02434-t001:** Impurities content in Ni foils (ppm).

Si	C	O	N	Mg	Sb	Fe	Mg	P
10	10	20	20	10	10	10	10	10

**Table 2 materials-16-02434-t002:** Impurities content in Ti foils (ppm).

Si	C	O	N	W	Ta	Fe	Nb
10	10	20	20	10	10	10	10

**Table 3 materials-16-02434-t003:** Impurities content in V foils (ppm).

Cr	C	O	Fe	Cd	Al	Mo	Mn	Ni	N	S	P	Co	Nb	Pb	Ti
27	160	140	35	20	220	60	5	10	10	25	24	10	7	5	10

**Table 4 materials-16-02434-t004:** Basic physical properties of Ni, Ti, Cr, and V (https://pt.kle.cz/zh_CN/index.html, accessed on 15 March 2023).

Metal	Lattice	Density/g·cm^−3^	Melting Point/°C	Thermal Conductivity/W·(m∙K)^−1^	HV/MPa
Ni	FCC	8.91	1455	90.7	638
Ti	HCP	4.51	1660	21.9	970
Cr	BCC	7.19	1857	93.7	1060
V	BCC	6.11	1900	30.7	628

The above data are parameters at room temperature except for melting point.

**Table 5 materials-16-02434-t005:** Three-point bending properties of W/M_nm_/TiN/Ta composites as SPSed at 1400 °C.

Composites	Flexural Stress/MPa	Bending Strain/%	Integration/kJ∙m^−2^
W/TiN/Ta	886 ± 5	0.90 ± 0.05	313.9
W/Ni_nm_/TiN/Ta	1150 ± 30	1.10 ± 0.1	261.6
W/Ti_nm_/TiN/Ta	879 ± 7	0.60 ± 0.1	273.6
W/Cr_nm_/TiN/Ta	912 ± 37	0.80 ± 0.15	201.9

**Table 6 materials-16-02434-t006:** Three-point bending properties of W/M/TiN/Ta composites as SPSed at 1200 °C.

Composites	Flexural Stress/MPa	Bending Strain/%	Integration/kJ∙m^−2^
W/Ta	1041 ± 13	1.00 ± 0.1	325.1
W/Ni/TiN/Ta	891 ± 26	0.60 ± 0.1	199.5
W/Ti/TiN/Ta	922 ± 18	1.20 ± 0.3	314.2
W/V/TiN/Ta	1294 ± 14	1.00 ± 0.15	419.3

**Table 7 materials-16-02434-t007:** Three-point bending properties of W/M/TiN/Ta composites joined at 1400 °C.

Composites	Flexural Stress/MPa	Bending Strain/%	Integration/kJ∙m^−2^
W/Ta	733 ± 9	3.5 ± 0.3	689.4
W/TiN/Ta	886 ± 5	0.9 ± 0.05	313.9
W/Ti/TiN/Ta	679 ± 11	0.7 ± 0.1	342.4
W/V/TiN/Ta	1136 ± 20	1.3 ± 0.1	213.7
W/V/TiN/Ta (joined at 1600 °C)	926 ± 14	1.0 ± 0.15	127.9

**Table 8 materials-16-02434-t008:** δ and D values of parallel crystal planes in W/V/TiN/Ta composites.

Interplanar Distance/nm		Joining Temperature/°C	δ/%	D/nm
(01$\overset{-}{1}$)_Ta_, 0.2338	(200)_TiN_, 0.2119	1200	9.8	2.2740
($\overset{-}{2}20$)_V_, 0.1071	(220)_TiN_, 0.1498	1200	33.2	0.3869
($\overset{-}{1}10$)_V_, 0.2143	(220)_TiN_, 0.1498	1200	35.4	0.5143

## Data Availability

The data that support the findings of this study are available from the corresponding author upon reasonable request.

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
