# Peer review of "Effect of Transition Metal Layer on Bending and Interfacial Properties of W/TiN/Ta-Laminated Composite"

_materials, 2023, doi:10.3390/ma16062434_

Round 1

Reviewer 1 Report

In this paper, the work done to study the properties of W/M/TiN/Ta layered composite with transition layer M, prepared by SPS method. With the changing temperatures flexural strength and strain of the composite were varies accordingly.

As of the title, it requires modification: the correct term is “transition metal”. And the usage of “its mechanism” at the end, was sound incomplete.

The Abstract require further polishing. SPS full term was not being introduced first. As well as EPMA, TEM etc…. Language needs further check.

Result from the HRTEM was presented, but the details of the equipment used was not properly introduced.

Conclusion section require statement that relate back to the potential application of the materials.

Overall, the draft needs to be polished further for the language style. Error and syntax can be spotted at most place.

Reviewer 2 Report

The meaning of  abbreviations should be determined: SPS, EBE, EPMA, HADDF, EDS

1.  What is the main question addressed by the research?

Study the bonding among the layers of wolfram based composite, and improve the toughness properties of the W based composite.

2. Do you consider the topic original or relevant in the field? Does it

address a specific gap in the field?

Yes, it is original  and relevant.

3. What does it add to the subject area compared with other published

material?

It studies the transition layers of a W/M/TiN/Ta layered composites, very special

4. What specific improvements should the authors consider regarding the

methodology? What further controls should be considered?

The testing methodology is “traditional”, mechanical testing and electron microscope used. They are applied to qualify a developing material.

5. Are the conclusions consistent with the evidence and arguments presented

and do they address the main question posed? Yes.

6. Are the references appropriate? Yes

7. Please include any additional comments on the tables and figures.

Some of the figures are too small, loosing information.

Reviewer 3 Report

This paper studies W/TiN/Ta’s strength and plasticity with a metal transition layer M. The experiments are done in detail with HRTEM study, and the conclusion is supported. See the comments below:

1. There are two Figure 3.

2. Figure 15: what is the hole in the center of the TEM sample?

3. Figure 17: there is a hole between the Ti- and Ta- rich layers - does this imply non-ideal coherence of the interface that can result in failure during service? 

4. Figure 20 (a): The diffraction pattern label is incorrect, please check.

5. Please remove “Second item;” at the end of the first conclusion.

6. Table 5-7: how many tests were conducted for each material? What’s the error bar? There are two Table 6.
